

# How are functionally similar code clones syntactically different? An empirical study and a benchmark

Stefan Wagner, Asim Abdulkhaleq, Ivan Bogicevic, Jan-Peter Ostberg and Jasmin Ramadani

Institute of Software Technology, University of Stuttgart, Stuttgart, Germany

## ABSTRACT

**Background**. Today, redundancy in source code, so-called "clones" caused by copy &paste can be found reliably using clone detection tools. Redundancy can arise also independently, however, not caused by copy&paste. At present, it is not clear how only *functionally similar clones* (FSC) differ from clones created by copy&paste. Our aim is to understand and categorise the syntactical differences in FSCs that distinguish them from copy&paste clones in a way that helps clone detection research.

**Methods**. We conducted an experiment using known functionally similar programs in Java and C from coding contests. We analysed syntactic similarity with traditional detection tools and explored whether concolic clone detection can go beyond syntax. We ran all tools on 2,800 programs and manually categorised the differences in a random sample of 70 program pairs.

**Results**. We found no FSCs where complete files were syntactically similar. We could detect a syntactic similarity in a part of the files in <16% of the program pairs. Concolic detection found 1 of the FSCs. The differences between program pairs were in the categories algorithm, data structure, OO design, I/O and libraries. We selected 58 pairs for an openly accessible benchmark representing these categories.

**Discussion**. The majority of differences between functionally similar clones are beyond the capabilities of current clone detection approaches. Yet, our benchmark can help to drive further clone detection research.

## INTRODUCTION

As software is now a key ingredient of current complex systems, the size of software systems is continuously increasing. While software with a code size of several thousand lines has been considered large in the seventies and eighties, we now reach code sizes of tens to hundreds of millions of lines of code. This has a strong effect on the complexity and manageability of these systems and, as a result, on the cost of maintaining them.

By abstraction and code generation, modern programming languages and development techniques help to reduce the amount of code we have to understand. Nevertheless, it still tends to be overwhelming. A factor that aggravates the situation is that there is unnecessary code in these huge code bases: unreachable code, never executed code and redundant code. The latter has been of increasing interest in the software engineering

Corresponding author
Stefan Wagner,
stefan.wagner@informatik.uni-stuttgart.de

research community under the term "cloning." In particular, clones that resulted from copy&paste can now be detected reliably (*Rattan, Bhatia & Singh, 2013*). In our own research with companies, we often found rates of redundant code caused by copy&paste in the range of 20%–30% (*Wagner, 2013*).

More challenging is the detection of functionally similar source code. We will refer to it as *functionally similar clones* (FSCs). FSCs might not have been created by copy&paste but developers independently needed and implemented certain functionalities in their code base. We deliberately go beyond *type-4 clones* (*Koschke, 2007*) or *simions* (*Deissenboeck et al., 2012*) which only include functional equivalence. In a refactoring session with the goal to reduce the size of a code base, a developer would still be interested in mostly similar and not only exactly equivalent functionality. Although this problem is in general undecidable, there have been several heuristic efforts (*Marcus & Maletic, 2001*; *Jiang & Su, 2009*; *Deissenboeck et al., 2012*; *Kim et al., 2011*).

*Juergens, Deissenboeck & Hummel (2010b)* showed that traditional clone detection approaches and tools are hardly able to detect functionally equivalent clones because they rely on syntactic similarities.

## Problem statement

So far, the work by *Juergens, Deissenboeck & Hummel (2010b)* is the only study investigating the syntactical differences in functionally similar clones. Furthermore, their study is limited: they use only programs implementing a single specification in Java. Therefore, we have no clear understanding of what differences make an individually developed functionally similar clone really different from copy&paste clones. Hence, a realistic, open benchmark for comparing and improving such approaches is also lacking although it is necessary for faster progress in the field (*Lakhotia et al., 2003*).

## Research objectives

The objective of this study is to better understand the differences that make up functionally similar clones to support future research on their detection. In particular, we want to classify and rate differences and build a representative benchmark.

## Contribution

We contribute a large-scale quantitative study combined with a qualitative analysis of the differences. We selected 2,800 Java and C programs which are solutions to the Google Code Jam programming contest and are therefore functionally similar. We identified copy&paste clones by using two clone detection tools (ConQAT and Deckard) to quantify syntactic similarities. We explored how a type-4 detection tool (CCCD) using cocolic detection performs in detecting the not syntactically similar FSCs. We created a categorisation of differences between undetected clones and quantified these categories. Finally, we derived a benchmark based on real FSCs covering the categories and degrees of differences which can drive the improvement of clone detection tools.

## Terminology

The most important terminological issue for this study is whether we need a new term for the phenomenon we investigate. The most established term is *type-4 clone*. Yet, the

definition in *Roy, Cordy & Koschke (2009)* emphasises that the code fragments have to perform the *same* computation. We want to emphasise the *similarity*, however. This is expressed by the term *simion*, introduced in *Juergens, Deissenboeck & Hummel (2010b)*, but that term has not been used in other context. Therefore, we propose the more general term *functionally similar clone*. Nevertheless, we believe the community should agree on one of these terms to avoid confusion in the future.

We define *functionally similar clones* as *two code fragments that provide a similar functionality w.r.t a given definition of similarity but can be implemented quite differently.* With this definition, we are based on the general definition of a code clone: "two code fragments form a clone pair if they are similar enough according to a given definition of similarity" (*Bellon et al., 2007*). Intuitively, what we are interested in is similar enough so that the clones are interesting for a developer of the system while changing it. For our FSCs, we expect this to include similarity in terms of the same output on the same input in *most* cases. The data types of the input and output should be somehow transformable into each other. The nonfunctional properties of the code fragments, however, can be very different.

For example, as a developer, I would be interested in the different implementations of search algorithms in my system also when they have quite different performance characteristics. I could then decide if the used algorithms and implementations are suitable for the context. Finally, as with other code clones, the similarity should be such that the clones could potentially be refactored. If the fragments are so different that a refactoring is very elaborate or results in a very complex design, a developer will not be interested in it.

As there is a large diversity in how the further terms around FSCs are used, we provide definitions for the clone types we investigate in this paper. Moreover, we define terms for granularities of the software programs under analysis in Table 1.

The structure of the remainder of the paper follows the guidelines in *Jedlitschka & Pfahl (2005)*.

## RELATED WORK

A *code clone* consists of at least two pieces of code that are similar according to a definition of similarity. Most commonly, **clone detection** approaches look for exact clones (also called *type-1*) and clones with simple changes such as renaming (also called *type-2*). These types of clones are detectable today in an efficient and effective way. Even clones with additional changes (*inconsistent*, *near-miss* or *type-3* clones) can be detected by several detection approaches and tools (*Kamiya, Kusumoto & Inoue, 2002*; *Deissenboeck et al., 2008*; *Jiang et al., 2007a*). There are also two surveys (*Koschke, 2007*; *Roy & Cordy, 2007*) and a systematic literature review (*Rattan, Bhatia & Singh, 2013*) on this topic. *Tiarks, Koschke & Falke (2011)* investigated in particular type-3 clones and also their differences. They concentrated, however, on differences in code metrics (e.g., fragment size), low level edits (e.g., variable) and abstracted them only slightly (e.g., to type substitution).

*Juergens, Deissenboeck & Hummel (2010b)* reported on an experiment to investigate the differences between syntactical/representational and semantic/behavioural similarities of

**Table 1  Terminology.**

| | |
|---|---|
| Type-1 clone | Similar code fragments except for variation in whitespace, layout and comments (*Bellon et al., 2007*) |
| Type-2 clone | Similar code fragments except for variation in identifiers, literals, types, whitespaces layouts and comments (*Bellon et al., 2007*) |
| Type-3 clone | Similar code fragments except that some statements may be added or deleted in addition to variation in identifiers, literals, types, whitespaces, layouts or comments (*Bellon et al., 2007*) |
| Type-4 clone | Two or more code fragments that perform the same computation but are implemented by different syntactic variants. (*Roy, Cordy & Koschke, 2009*) |
| Functionally similar clone (FSC) | Code fragments that provide a similar functionality w.r.t a given definition of similarity but can be implemented quite differently |
| Solution file | A single program in one file implementing the solution to a programming problem |
| Solution set | A set of solution files all solving the same programming problem |
| Clone pair | Two solution files from the same solution set which we assume to be functionally similar |

code and the detectability of these similarities. They used a simple student assignment called *email address validator* and also inspect the open-source software *JabRef*. Both of them are in Java. To detect the clones of types 1–3, they used the clone detection tools ConQAT and Deckard. They reviewed the open-source system manually to identify if behaviourally similar code that does not result from copy&paste can be detected and occurs in real-world software. The results indicate that behaviourally similar code of independent origin is highly unlikely to be syntactically similar. They also reported that the existing clone detection approaches cannot identify more than 1% of such redundancy. We build our work on their study but concentrate on understanding the differences in more detail based on a diverse sample with a larger sample size and different programming languages.

Several researchers have proposed to move away from the concrete syntax to detect what they call **semantic clones**. *Marcus & Maletic (2001)* used information retrieval techniques on source code to detect semantic similarities. *Krinke (2001)* proposed to use program dependence graphs (PDG) for abstracting source code. *Higo et al. (2011)* extended this to an incremental approach. *Komondoor & Horwitz (2001)* also used PDGs for clone detection and see the possibility to find non-contiguous clones as a main benefit. *Gabel, Jiang & Su (2008)* combined the analysis of dependence graphs with abstract syntax trees in the tool Deckard to better scale the approach.

A very different approach to detecting semantic clones comes from *Kim et al. (2011)* who use static analysis to extract the memory states for each procedure exit point. They could show that they find more semantically similar procedures as clones than previous clone detectors including PDG-based detectors. Finally, *Kamiya (2013)* proposed to analyse potential executions by analysing method calls on the Bytecode level in Java programs.

Nevertheless, the used approach as well as the examples of found semantic clones suggest that the syntactic representation still plays a role and that the clones have been created by copy&paste. These semantic clone detection techniques cannot guarantee that they also find all functionally similar clones as a completely different structure and memory states can generate similar functionality.

*Jiang & Su (2009)* were the first to comprehensively detect **functionally similar code** by using random tests and comparing the output. Hence, they were also the first who were able to detect clones without any syntactic similarity. They claim they are able to detect "*functionally equivalent* code fragments, where functional equivalence is a particular case of semantic equivalence that concerns the input/output behavior of a piece of code." They detected a high number of functionally equivalent clones in a sorting benchmark and the Linux kernel. Several of the detected clones are dubious, however, as it is not clear how useful they are. They state: "Assuming the input and output variables identified by EQMINER for these code fragments are appropriate, such code fragments are indeed functionally equivalent according to our definition. However, whether it is really useful to consider them functionally equivalent is still a question worth of future investigation."

*Deissenboeck et al. (2012)* followed an analogous approach to *Jiang & Su (2009)* to detect functionally similar code fragments in Java systems based on the fundamental heuristic that two functionally similar code fragments will produce the same output for the same randomly generated input. They implemented a prototype based on their toolkit ConQAT. The evaluation of the approach involved 5 open-source systems and an artificial system with independent implementations of the same specification in Java. They experienced low detection results due to the limited capability of the random testing approach. Furthermore, they mention that the similarities are missed due to chunking, i.e., if the code fragments perform a similar computation but use different data structures at their interfaces. They emphasise that further research is required to understand these issues.

CCCD (*Krutz & Shihab, 2013*) also claims to detect functionally similar code for C programs based on concolic analysis. Its creators evaluated their implementation of the approach on the benchmarks mentioned below and found a 92% recall even in the type-4 clones in those benchmarks. As the tool is freely available in a virtual machine, we were able to include it in our experiment.

A clear comparison and measurement of the improvement in clone detection research would require a comprehensive **benchmark**. There have been few approaches (*Lakhotia et al., 2003*; *Roy, Cordy & Koschke, 2009*; *Tempero, 2013*) trying to establish a benchmark but they are either small and artificial or do not contain (known) FSCs. The only exception is the recent *BigCloneBench* (*Svajlenko et al., 2014*) which has a huge number of clones mined from source code repositories. Yet, they do not classify the types of differences and also state "there is no consensus on the minimum similarity of a Type-3 clone, so it is difficult to separate the Type-3 and Type-4 clones".

## EXPERIMENTAL DESIGN

To reach our research objectives, we developed a study design based on the idea that we investigate sets of programs which we knew to be functionally similar: accepted submissions

to programming contests. We formulated four research questions which we all answer by analysing these programs and corresponding detection results. All instrumentation, analysis scripts and results are freely available in *Wagner et al. (2014)*.

### Research questions

As we have independently developed but functionally similar programs, we first wanted to establish how much syntactic similarity is in these programs. We can investigate this by quantifying the share of type-1–3 clones

**RQ 1: What share of independently developed similar programs are type-1–3 clones?**

   Then we wanted to understand what is different in clones not of type-1–3. This should result in a categorisation and rating of the differences between FSCs.

**RQ 2: What are the differences between FSC that go beyond type-1–3 clones?**

   Although we could not fully evaluate type-4 detectors, we wanted at least to explore what a modern clone detection approach can achieve on our FSCs. This should give us an indication how much more research is needed on those detection approaches.

**RQ 3: What share of FSC can be detected by a type-4 clone detector?**

   Finally, to make our results an operational help for clone detection research, we wanted to create a representative benchmark from the non-detected clones.

**RQ 4: What should a benchmark contain that represents the differences between FSC?**

### Hypotheses

We define two hypotheses regarding RQ 1. As we investigate the share of detectable Type-1–3 clones, we wanted to understand if there are differences between the used tools and analysed languages because this might have an influence on the generalisability of our results. We formulated the two null hypotheses:

**H1: There is no difference in the share of detected Type-1–3 clones between the analysed programming languages.**

**H2: There is no difference in the share of detected Type-1–3 clones between the analysed clone detection tools.**

   Moreover, in RQ 2, we wanted to understand the characteristics of non-detected clone pairs and, therefore, categorised them. In this categorisation, we also rated the degree of difference in each category. An ideal categorisation would have fully orthogonal categories and, hence, categories would not be correlated in the degree of difference:

**H3: There is no correlation between the degrees of difference between categories.**

   Furthermore, we could imagine that different programming languages might cause disparately strong differences in certain categories. As this again has an impact on the generalisability of our results, we formulated this null hypotheses:

**H4: There is no difference in the degree of difference between the analysed programming languages.**

### Design

The overall study design is a combination of quantitative and qualitative analysis. For the quantitative part of our study, we used a factorial design with two factors (programming language and clone detection tool). As applying the treatments of both factors was mostly

**Table 2** The factorial design used in this experiment.

| | | Programming language | |
| | | Java | C |
| --- | --- | --- | --- |
| Clone | CCCD | – | ✓ |
| Detection | ConQAT | ✓ | ✓ |
| Tool | Deckard | ✓ | ✓ |

automated we could apply almost all factor levels to all study object programs (which we call *solutions*). Only if a detection tool did not support a certain programming language, we would not apply it. We tried to minimise that but to include a contemporary tool, we accepted an unbalanced design. Table 2 shows the factors in our experiment.

We will describe the programming languages, clone detection tools and corresponding programs under analysis in more detail in the next subsection.

## Objects

The general idea of this experiment was that we analyse accepted solutions to programming contests because we know that for a given problem, the solutions must be functionally similar. Therefore, our selection of study objects needed to include clone detection tools we could access and execute as well as solutions in programming languages supported by most of the detection tools.

### *Clone detection tools*

Primarily, we needed clone detection tools for detecting type-1–3 clones to investigate with RQ 1 the syntactic similarity of FSCs. We did a literature and web search for available tools.

Many research prototypes were not available or could not be brought to execute correctly. Several tools were not included in the study due their lower performance and scalability or their lack of support for some clone types. CloneDR and CPMiner have lower performance and scalability compared to Deckard (*Jiang et al. 2007a*). CCFinder has also lower performance than Deckard and does not support type-3 clones (*Svajlenko & Roy, 2014*).

In the end, we chose two clone detection tools that both can analyse Java and C programs: **ConQAT** (*Deissenboeck et al., 2008*) and **Deckard** (*Jiang et al., 2007a*). ConQAT is described in *Rattan, Bhatia & Singh (2013)* as modern, useful and fast open-source clone detector framework. In the studies we mentioned above, Deckard has shown to have good performance and scalability. Hence, both are well established and have been used in previous studies, especially *Juergens, Deissenboeck & Hummel (2010b)*. At the time of the study, those were the two tools which were both freely available and were possible to make them work for us.

**ConQAT** is a stable open-source dashboard toolkit also used in industry. It is a general-purpose tool for various kinds of code measurement and analysis. For our experiment, ConQAT offers several specific clone detection configurations for various programming languages including Java, C/C++, C# and Cobol. It has separate detection algorithms

for type-1/2 clones and type-3 clones. We employed the latter algorithm. ConQAT has been used in various studies on clone detection (*Juergens et al., 2009*; *Juergens et al., 2010a*) including the study we build on *Juergens, Deissenboeck & Hummel (2010b)*.

The language-independent clone detection tool **Deckard** works on code in any programming language that has a context-free grammar. Deckard uses an efficient algorithm for identifying similar subtrees and applies it to tree representations of source code. It automatically generates a parse tree builder to build parse trees required by its algorithm. By a similarity parameter it is possible to control whether only type-1/2 clones or type-3 clones are detected. Deckard is a stable tool used in other studies (*Gabel, Jiang & Su, 2008*; *Jiang, Su & Chiu, 2007b*) including the study we build on.

To explore the state of type-4 clone detection tools, we also searched for such tools. Most existing tools, however, could not be used. For example, EqMiner (*Jiang & Su, 2009*) was too tightly coupled with the Linux kernel and MeCC (*Kim et al., 2011*) could not detect clones across files. Finally, we were able to only include a single type-4 detector.

**CCCD** (*Krutz & Shihab, 2013*) is a novel clone detection tool that uses concolic analysis as its primary approach to detect code clones. Concolic analysis combines symbolic execution and testing. CCCD detects only clones in programs implemented in C. The concolic analysis allows CCCD to focus on the functionality of a program rather than the syntactic properties. Yet, it has the restriction that it only detects function-level clones.

### Solution sets and solutions

We looked at several programming contests and the availability of the submitted solutions. We found that *Google Code Jam* (https://code.google.com/codejam/) provided us with the broadest selection of programming languages and the highest numbers of submissions. Google Code Jam is an annual coding contest organised by Google. Several tens of thousands of people participate each year. In seven competition rounds, the programmers have to solve small algorithmic problems within a defined time frame. Although over one hundred different programming languages are used, the majority of the solutions are in C, C++, Java and Python. Most solutions of the participants are freely available on the web (http://www.go-hero.net/jam/14/).

We define a *solution* as a single code file delivered by one participant during the contest. We define a *solution set* as a set of solutions all solving the same problem. A typical solution set consists of several hundred to several thousand solutions. We can be sure that all solutions in a solution set should be FSCs because they passed the judgement of the programming contest. This is also implemented as automated tests (https://code.google.com/codejam/quickstart.html). Even if there are differences in the programs, e.g., in the result representation, these are instances of similarity instead of equivalence.

We selected 14 out of 27 problem statements of the Google Code Jam 2014. For every problem we randomly chose 100 solutions in Java and 100 solutions in C from sets of several hundreds to several thousands of solutions. Table 3 shows a summary of the size of the chosen solution sets. Hence, on average a C solution has a length of 46 LOC and a Java solution of 94 LOC.

**Table 3  Summary of the solution sets.**

|  | #No. sets | #Files/set | #Procedures | LOC |
|---|---|---|---|---|
| C | 14 | 100 | 2,908 | 64,826 |
| Java | 14 | 100 | 8,303 | 131,398 |

**Table 4  Information on the solution sets.**

| Java | | | | C | | | |
|---|---|---|---|---|---|---|---|
| Set | #Files | LOC | #Proc. | Set | #Files | LOC | #Proc. |
| 1 | 100 | 11,366 | 823 | 1 | 100 | 3,917 | 233 |
| 2 | 100 | 7,825 | 523 | 2 | 100 | 3,706 | 167 |
| 3 | 100 | 10,624 | 575 | 3 | 100 | 4,750 | 265 |
| 4 | 100 | 6,766 | 473 | 4 | 100 | 3,928 | 219 |
| 5 | 100 | 7,986 | 585 | 5 | 100 | 4,067 | 187 |
| 6 | 100 | 10,137 | 611 | 6 | 100 | 6,840 | 166 |
| 7 | 100 | 13,300 | 869 | 7 | 100 | 4,701 | 263 |
| 8 | 100 | 8,568 | 614 | 8 | 100 | 4,679 | 176 |
| 9 | 100 | 8,580 | 717 | 9 | 100 | 6,831 | 227 |
| 10 | 100 | 9,092 | 459 | 10 | 100 | 4,063 | 159 |
| 11 | 100 | 8,536 | 584 | 11 | 100 | 4,624 | 266 |
| 12 | 100 | 11,412 | 648 | 12 | 100 | 3,574 | 163 |
| 13 | 100 | 9,436 | 465 | 13 | 100 | 3,335 | 168 |
| 14 | 100 | 7,770 | 357 | 14 | 100 | 5,811 | 249 |

In Table 4, we detail the size of the selected solution sets. The solution sets differ in size but the means all lie between 33 and 133 LOC per solution.

## Data collection procedure
### Preparation of programs under analysis

We implemented an instrumentation which automatically downloaded the solutions from the website, sampled the solution sets and solutions and normalised the file names. The instrumentation is freely available as Java programs in *Wagner et al. (2014)*. Every downloaded solution consisted of a single source code file.

### Configuration of clone detection tools

We installed ConQAT, Deckard and CCCD and configured the tools with a common set of parameters. As far as the parameters between the tools were related to each other, we tried to set the same values based on the configuration in *Juergens, Deissenboeck & Hummel (2010b)*. We set the parameters conservatively so that the tools find potentially more clones as we would normally consider valid clones. This ensured that we do not reject our null hypotheses because of configurations. We based the minimal clone length on the parameters for practical use mentioned in *Juergens, Deissenboeck & Hummel (2010b)* and experimented with lower values. Finally, we decide to make 6 the threshold, because

lower values created many findings which we did not consider a valid clone. For the gap clone ratio, we also tried different values, but decided to use 0.3 as it was close to the recommended value, generated reasonable clones and clones similar to the ones detected by CCCD with the Levenshtein score of 35. All the detailed configurations are available in *Wagner et al. (2014)*.

For CCCD, we only included clone pairs which have a Levenshtein similarity score below 35 as advised in *Krutz & Shihab (2013)* where the score is calculated for the concolic output for each function. The detection failed for 4 of the solutions of our sample set. We had to exclude them from further analysis.

### Executing clone detection tools

We manually executed the clone detection tools for every solution set. ConQAT generated an XML file for every solution set containing a list of found clone classes and clones. Deckard and CCCD generate similar CSV files. Our instrumentation tool parsed all these result files and generated reports in a unified format. The reports are tables in which both rows and columns represent the solutions. The content of the table shows the lowest detected clone type between two files. Additionally, our tool calculated all shares of syntactic similarity as described in the next section and wrote the values into several CSV files for further statistical analysis. We also wrote all the detected clones into several large CSV files. Altogether, the tools reported more than 9,300 clones within the Java solutions and more than 22,400 clones within the C solutions.

## Analysis procedure
### Share of syntactic similarity (RQ 1)

All solutions in a solution set solve the same programming problem and were accepted by Google Code Jam. Hence, their functionality can only differ slightly and, therefore, they are functionally similar. To understand how much of this similarity is expressed in syntactic similarity, we calculate the share of FSCs which are also type-1–2 or type-1–3 clones.

Inspired by *Juergens, Deissenboeck & Hummel (2010b)*, we distinguish partial and full syntactic similarity. The *share of full syntactic similarity* is the ratio of clone pairs where all but a defined number of the statements of the solutions of the pair were detected as a clone in relation to all clone pairs. We set the threshold of this to a maximum of 16 lines of code difference within a clone pair, which leads to ratios of 5%–33% of difference based on the solutions' lines of code.

$$\text{Share of full syntactic similarity} = \frac{|\text{Found full clone pairs}|}{|\text{All clone pairs}|}. \tag{1}$$

Because we expected the share of full syntactic similarity to be low, we wanted to check whether there are at least some parts with syntactic similarity. It would give traditional clone detection tools a chance to hint at the FSC. Furthermore, it allowed us to inspect more closely later on what was not detected as a clone. We called the ratio *share of partial syntactic similarity*.

$$\text{Share of partial syntactic similarity} = \frac{|\text{Found partial clone pairs}|}{|\text{All clone pairs}|}. \tag{2}$$

For a more differentiated analysis, we calculated two different shares each representing certain types of clones. We first computed the share for type-1–2 clones. This means we only need to accept exact copies, reformatting and renaming. Then, we determined the shares for type-1–3 clones which includes type-1–2 and adds the additional capability to tolerate smaller changes.

In ConQAT and Deckard, we can differentiate between type-1/2 clones and type-3 clones by configuration or result, respectively. In ConQAT, clones with a gap of 0 are type-1/2 clones. In Deckard, analysis results with a similarity of 1 are type-1/2 clones. The others are type-3 clones. The instrument tooling described in 'Data collection procedure' directly calculated the various numbers. We computed means per clone type and programming language.

For a further statistical understanding and to answer the hypotheses H1–H4, we did statistical hypotheses tests. For answering H1 and H2, we performed an analysis of variance (ANOVA) on the recall data with the two factors *programming language* and *detection tool*. We tested the hypotheses at the 0.05 level. All analyses implemented in R together with the data are available in *Wagner et al. (2014)*.

The combined descriptive statistics and hypothesis testing results answered RQ 1.

### Classifying differences (RQ 2)

For the categorisation of the differences of FSCs that were not syntactically similar, we took a random sample of these clone pairs. As we had overall 69,300 clone pairs for Java and C, we needed to restrict the sample for a manual analysis. We found in an initial classification (see also 'Validity procedure') that a sample of 0.5‰ per language and fully/partially different clone pairs is sufficient for finding repeating categories and getting a quantitative impression of the numbers of clone pairs in each category. With larger samples, the categories just kept repeating. Therefore, we took a sample of 2‰ of the syntactically different clone pairs: 70 pairs each of the fully and partially different clone pairs (35 C and 35 Java).

The set of *fully syntactically different clone pairs* is the set of all pairs in all solution sets minus any pair detected by any of the type-1–3 detection. We apply random sampling to get pairs for further analysis: first, we randomly selected one of the solution sets in a language. Second, we randomly selected a solution file in the solution set and checked if it was detected by Deckard or ConQAT. If it was detected, we would discard it and select a new one. Third, we randomly picked a second solution file, checked again if it was detected and discard it if it was.

The set of *partially syntactically different clone pairs* is then the superset of all partially different clone pairs minus the superset of all fully different clone pairs. From that set, we randomly selected clone pairs from all partially different pairs of a programming language and checked if it was fully different. If that was the case, we would discard it and take a new random pair. We found their analysis to be useful to understand also smaller syntactic differences.

We then employed qualitative analysis. We manually classified the characteristics in the clone pairs that differed and, thereby, led to being not detected as type-1–3 clone. This

classification work was done in pairs of researchers in three day-long workshops in the same room. It helped us to discuss the categories and keep them consistent. The result is a set of categories of characteristics that describe the differences. We added quantitative analysis to it by also counting how many of the sampled clone pairs have characteristics of the found types.

After the creation of the categories we also assessed the degree of difference (high, medium, or low) per category. From the discussion of the categories, we discovered that this gave us a comprehensive yet precise way to assign clone pairs to the categories. Furthermore, it gave us additional possibilities for a quantified analysis. First, we wanted to understand better how we categorised and assessed the degrees of difference as well as answer H3. Therefore, we performed correlation analysis on them. We chose Kendall's tau as the correlation coefficient and tested all correlations on the 0.05 level.

For answering H4, we performed a multivariate analysis of variance (MANOVA) which allows more than one dependent variable to be used. Here, our dependent variables are the degrees of difference and the independent variable is the programming language. In this analysis, we have a balanced design because we ignored the category *OO design* which was only applicable to Java programs. We use the Pillar-Bartlett statistic for evaluating statistical significance. We checked H4 also on the 0.05 level.

These categories with frequencies as well as the results of the hypothesis tests answered RQ 2.

### Running a type-4 detector (RQ 3)

As this part of the study is only for exploratory purposes, we focused on the recall of CCCD in the FSCs. As all solutions contain a *main* function, we expected it to find each *main*-pair as clone. We calculate the recall as the number of detected clone pairs by the sum of all clone pairs. A perfect clone detection tool would detect all solutions from a solution set as clones.

### Creating a benchmark (RQ 4)

After the categorisation to answer RQ 2, we had a clear picture of the various differences. Therefore, we could select representative examples of each difference for each programming language and put them into our new benchmark. To check that the clone pairs cannot be detected by the tools, we run the tools again on the benchmark. If one of the tools still detected a clone, we would replace the clone pair by another representative example until no clones are detected.

We created the benchmark by choosing clone pairs consisting of two source code files out of the same solution set. The two files therefore solve the same problem. We selected three pairs where the difference between the files belong to that category for each of the categories we created by answering RQ 2. We chose three pairs for all of the three levels of difference. The other categories of the pairs are low or non-existent so that the pair is focused on one category. Additionally, we added one extra clone pair with extreme differences in all categories.

Preferably, we would provide the source code of the chosen solutions directly all in one place. Yet, the copyright of these solutions remains with their authors. Therefore, we

**Table 5** Full and partial syntactic similarity (in %).

| | | Partially similar | | | | Fully similar | | | |
|---|---|---|---|---|---|---|---|---|---|
| | | Type 1–2 | | Type 1–3 | | Type 1–2 | | Type 1–3 | |
| Lang. | Tool | Mean | SD | Mean | SD | Mean | SD | Mean | SD |
| Java | ConQAT | 6.36 | 0.05 | 11.53 | 0.07 | 0.00 | 0.00 | 0.00 | 0.00 |
| | Deckard | 0.33 | 0.00 | 0.87 | 0.01 | 0.00 | 0.00 | 0.00 | 0.00 |
| | **Mean** | 3.35 | 0.03 | 6.11 | 0.04 | 0.00 | 0.00 | 0.00 | 0.00 |
| C | ConQAT | 5.24 | 0.09 | 11.48 | 0.13 | 1.30 | 0.00 | 1.73 | 0.00 |
| | Deckard | 0.28 | 0.00 | 1.44 | 0.01 | 0.01 | 0.00 | 0.01 | 0.00 |
| | **Mean** | 1.82 | 0.00 | 4.32 | 0.06 | 0.47 | 0.00 | 0.58 | 0.00 |
| **Grand mean** | | 2.45 | 0.04 | 5.07 | 0.04 | 0.26 | 0.00 | 0.35 | 0.00 |

provide source files following the same structure as the original files but not violating the copyright.

A final set of clone pairs that are not detected as full clones by any of the tools constitutes the benchmark and answered RQ 4.

### Validity procedure

To avoid selection bias, we performed random sampling where possible. We randomly selected the solution sets and solutions that we use as study objects. In addition, before we manually analysed the category of syntactically different clone pairs, we chose random samples of clone pairs.

To avoid errors in our results, we manually checked for false positives and clone types with samples of clones in the solution sets. Furthermore, by working in pairs during all manual work, we controlled each other and detected problems quickly. Overall, the manual inspection of 70 clone pairs for RQ 2 also was a means to detect problems in the detection tools or our instrumentation.

For the manual categorisation, we started by categorising 30 syntactically different clone pairs to freely create the categories of undetected clone pairs. Afterwards, we discussed the results among all researchers to come to a unified and agreed categorisation. The actual categorisation of clone pairs was then performed on a fresh sample. Additionally, we performed an independent categorisation of a sample of 10 categorised clone pairs and calculated the inter-rater agreement using Cohen's kappa.

## ANALYSIS AND RESULTS

We structure the analysis and results along our research questions. All quantitative and qualitative results are also available in *Wagner et al. (2014)*.

### Share of syntactic similarity (RQ 1)

We summarised the results of the calculated shares for fully and partially syntactically similar clone pairs in Table 5. We divided the results by programming languages, detection tools and detected clone types. The results differ quite strongly from tool to tool but only

**Table 6  ANOVA results for variation in recalls (Type II sum of squares, * denotes a significant result).**

|  | Sum of squares | F value | Pr(>F) | |
| --- | --- | --- | --- | --- |
| Partial type 1–2 | | | | |
| Language | 0.0005 | 0.2352 | 0.6294 | |
| Tool | 0.0491 | 12.2603 | $3 \cdot 10^{-5}$ | * |
| Partial type 1–3 | | | | |
| Language | 0.0010 | 0.0210 | 0.8853 | |
| Tool | 0.1884 | 20.5846 | $1 \cdot 10^{-7}$ | * |
| Full type 1–2 | | | | |
| Language | $1 \cdot 10^{-7}$ | 7.8185 | 0.0072 | * |
| Tool | $2 \cdot 10^{-8}$ | 1.1566 | 0.2871 | |
| Full type 1–3 | | | | |
| Language | $2 \cdot 10^{-7}$ | 7.7757 | 0.0074 | * |
| Tool | $5 \cdot 10^{-8}$ | 1.9439 | 0.1692 | |

slightly between the programming languages. The average syntactic similarities and the standard deviations (SD) are all very low. ConQAT detects more full and partial clones in clone pairs.

Table 6 shows the ANOVA results which we need for answering hypotheses H1 and H2. As our experiment is unbalanced, we use the Type II sum of squares. This is possible because we found no significant interactions between the factors in any of the ANOVA results.

The results give us no single evaluation of the hypotheses H1 and H2. We have to differentiate between partial and full syntactic similarity. For the partial similarity, we consistently see a significant difference in the variation in the detection tools but not in the programming languages. Hence, for partial clone similarity, we corroborate H1 that there is no difference in recall between the analysed programming languages. Yet, we reject H2 in favour of the alternative hypothesis that there is a difference in the similarity share between different tools. For full similarity, we reject H1 in favour of the alternative hypothesis that there is a difference between the analysed programming languages. Instead, we accept H2 that there is no difference between the analysed detection tools.

How can we interpret these results? *The overall interpretation is that the share of syntactic similarity in FSCs is very small.* There seem to be many possibilities to implement a solution for the same problem with very different syntactic characteristics. When we only look at the full syntactic similarity, the results are negligible. Both tools detect none in Java and only a few clone pairs for C. Hence, the difference between the tools is marginal. The difference is significant between C and Java, however, because we found no full clone pairs in Java. As we saw in a manual inspection, the full detection is easier in C if the developers implement the whole solution in one main function.

For partial syntactic similarity, we get higher results but still stay below 12%. Hence, for almost 90% of the clone pairs, we do not even detect smaller similarities. We have no significant difference between the languages but the tools. ConQAT has far higher results

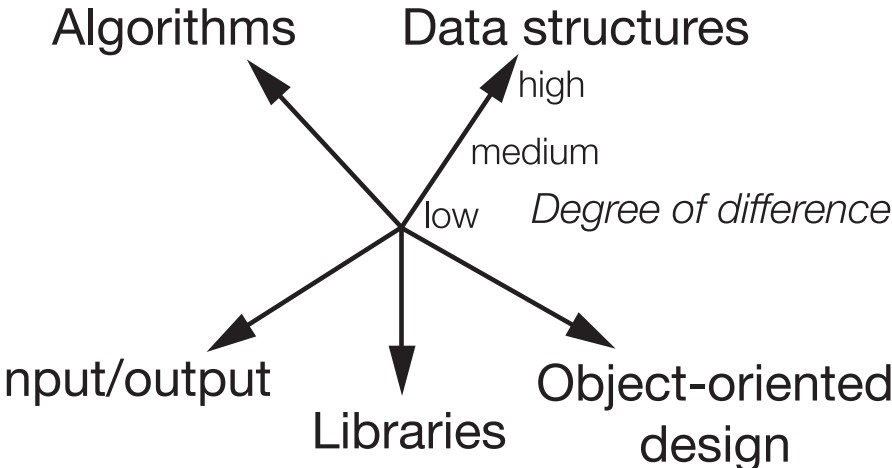

**Figure 1** The categories of characteristics of differences between clone pairs.

than Deckard in the type-1–3 clones. The distinct detection algorithms seem to make a difference here. For the further analysis, we accept an FSC as syntactically similar if one of the tools detected it.

## Categories of differences (RQ 2)

Initially, we created 18 detailed categories. In our qualitative analysis and discussions, we finally reduced them to five main categories of characteristics describing the differences between the solutions in a clone pair. The five categories are *algorithm*, *data structure*, *object-oriented design*, *input/output* and *libraries*. We could assign each of the 18 initial categories there and realised that we can assign them to different *degrees of difference*. Therefore, we ended up with a categorisation including an ordinal quantification of the degree of difference with the levels *low*, *medium* and *high*. The overall categorisation is shown in Fig. 1. The centre of the dimensions would be a type-1 clone. The further out we go on each dimension, the larger the difference.

To make the categories and degrees of difference clearer, we give examples of characteristics in clone pairs in Table 7 that led us to classify them in the specific degrees of difference. The guiding principle was how much effort (in terms of edit operations) it would be to get from one solution to the other.

The two main aspects in any program are its *algorithms* and its *data structures*. This is reflected in our two main categories. Our corresponding degrees of difference reflect that there might be algorithms that are almost identical with e.g., only a *switch* instead of nested *if* statements up to completely different solutions, e.g., iterative vs. recursive. Similiarly, in data structures, we can have very simple type substitutions which change the behaviour but are still functionally very similar (e.g., from *int* to *long*) but also completely user-defined data types with strong differences.

Related to data structures is the category *OO design*. We made this a separate category because it only applies to OO languages and it had a particular kind of occurrence in the

**Table 7    Examples for the levels in the degree of difference per category.**

| Algorithm | low | Only syntactic variations |
|---|---|---|
| | medium | Similarity in the control structure but different method structure |
| | high | No similarity |
| **Data structure** | low | Different data types, e.g., int—long |
| | medium | Related data types with different interface, e.g., array vs. List |
| | high | Standard data types vs. own data classes or structs |
| **OO design** | low | Only one/few static methods vs. object creation |
| | medium | Only one/few static methods vs. data classes or several methods |
| | high | Only one/few static methods vs. several classes with methods |
| **Library** | low | Different imported/included but not used libraries |
| | medium | Few different libraries or static vs. non-static import |
| | high | Many different or strongly different libraries |
| **I/O** | low | Writing to file vs. console with similar library |
| | medium | Strongly different library, e.g., Scanner vs. FileReader |
| | high | Strongly different library and writing to file vs. console |

programs we inspected. Some developers tended to write Java programs as if there were no object-oriented features, while others created several classes and used their objects.

As our programming environments and languages are more and more defined by available *libraries*, this was also reflected in the differences between solutions. If one developer of a solution knew about a library with existing functionality needed, and the other developer implemented it herself, this created code that looks strongly different but can have similar functionality.

Finally, maybe a category that arose because the programming contest did not specify if the input and output should come from a console or a file was the usage of *I/O*. Nevertheless, we think that this might also be transferable to other FSCs and contexts because we might be interested in functionally similar code even if one program writes the output on a network socket while the other writes into a file.

Table 8 shows descriptive statistics for the categories in our sample of undetected clone pairs. The column *Share* shows the ratio of clone pairs with a degree of difference higher than 0 in relation to all clone pairs in that language. The median and median absolute deviation (MAD) give the central tendency and dispersion of the degrees in that category. For that, we encoded *no difference* $= 0$, *low* $= 1$, *medium* $= 2$ and *high* $= 3$.

All categories occur in the majority of clone pairs. The categories *algorithm* and *libraries* even occur in about three quarters of the clone pairs. The occurrence of categories is consistently smaller in C than in Java. The medians are mostly low but with a rather large deviation. Only *input/output* in C has a median of 0. This is consistent with our observation during the manual inspection that I/O is done similarly in the C programs.

For evaluating H3, we calculated Kendall's correlation coefficients for all combinations of categories. The results are shown in Table 9. The statistical tests for these correlations

**Table 8   Descriptive statistics of degrees of difference over categories and programming languages.**

| Lang. | Category | Share | Median | MAD |
|---|---|---|---|---|
| Java | Algorithm | 96% | 3 | 0.0 |
| | Libraries | 86% | 1 | 1.5 |
| | I/O | 83% | 2 | 1.5 |
| | Data structure | 72% | 1 | 1.5 |
| | OO design | 71% | 1 | 1.5 |
| C | Algorithm | 76% | 2 | 1.5 |
| | Libraries | 73% | 1 | 1.5 |
| | Data structure | 66% | 1 | 1.5 |
| | I/O | 38% | 0 | 0.0 |
| Total | Algorithm | 86% | 2 | 1.5 |
| | Libraries | 79% | 1 | 1.5 |
| | OO design | 71% | 1 | 1.5 |
| | Data structure | 69% | 1 | 1.5 |
| | I/O | 60% | 1 | 1.5 |

**Table 9   Correlation matrix with Kendall's correlation coefficient for the category degrees (all are significant).**

| | Algo. | Data struct. | OO design | I/O | Libraries |
|---|---|---|---|---|---|
| Algorithm | 1.00 | 0.38 | 0.44 | 0.15 | 0.31 |
| Data struct. | 0.38 | 1.00 | 0.26 | 0.25 | 0.21 |
| OO design | 0.44 | 0.26 | 1.00 | 0.29 | 0.39 |
| I/O | 0.15 | 0.25 | 0.29 | 1.00 | 0.27 |
| Libraries | 0.31 | 0.21 | 0.39 | 0.27 | 1.00 |

**Table 10   MANOVA results for variation in degree of differences (Type I sum of squares, * denotes a significant result).**

| | Pillai-Bartlett | Approx. $F$ | Pr($>F$) | |
|---|---|---|---|---|
| Language | 0.1513 | 6.0196 | 0.0002 | * |

showed significant results for all the coefficients. Therefore, we need to reject H3 in favour of the alternative hypothesis that there are correlations between the degrees of difference between different categories.

Finally, for evaluating H4, we show the results of the MANOVA in Table 10. We can reject H4 in favour of the alternative hypothesis that there is a difference between the degrees of difference between the analysed programming languages. This is consistent with the impression from the descriptive statistics in Table 8.

In summary, we interpret these results such that *there are differences in FSC pairs in their algorithms, data structures, input/output and used libraries. In Java, there are also differences in the object-oriented design.* On average, *these differences are mostly small but the variance*

| Table 11 | Full and partial clone recall means over solution sets for CCCD (in %). | |
| --- | --- | --- |
| | **Mean** | **SD** |
| Partial | 16.03 | 0.07 |
| Full | 0.10 | 0.00 |

*is high*. Hence, we believe that with advances in clone detectors for tolerating the smaller differences, there could be large progress in the detection of FSCs. Yet, there will still be many medium to large differences. We also saw that the programming languages vary in the characteristics of undetected difference. Therefore, it might be easier to overcome those differences in non-object-oriented languages, such as C, than in object-oriented languages which offer even more possibilities to express solutions for the same problem. Yet, we were impressed by the variety in implementing solutions in both languages during our manual inspections.

Our categories are significantly correlated with each other. This can mean that there might be other, independent categories with less correlation. Nevertheless, we believe the categories are useful because they describe major code aspects in a way that is intuitively understandable to most programmers. It would be difficult to avoid correlations altogether. For example, a vastly different data structure will always lead to a very different algorithm.

### Type-4 detection (RQ 3)

Table 11 shows the recall of fully and partially detected clone pairs in our sample. CCCD has a considerable recall for partial clones in the clone pairs of about 16%. It does, however, detect almost none of the clone pairs as full clones. The overlap with ConQAT and Deckard, and therefore type-1–3 clones, is tiny (0.05% of the recall).

We interpret this result such that also contemporary type-4 detection tools have still problems detecting real-world FSCs and to handle the differences we identified in RQ 2.

### Benchmark (RQ 4)

The number of study objects used in our analysis is quite high. As described above, we examined 1,400 Java files and 1,400 C files. For many demonstrations and clone detection tool analyses a much smaller file set is sufficient. We call this smaller set of files *benchmark*.

The first half of the benchmark we provide consists of 29 clone pairs. For Java, we include 16 clone pairs. The set of clone pairs we provide for C is structured in exactly the same way as the Java samples except that we do not have the three clone pairs that differ only in object-oriented design. Therefore, we do not have 16 samples here but 13 which make the 29 clone pairs for both languages.

Figure 2 shows a rating of an example clone pair in the benchmark set where the two files only differ significantly in the kind of input/output, but not in the other categories. Figure 3 shows how such a high difference in the category input/output could look like between two solutions in a solution set.

We provide this distribution of clone pairs for both partial clones and full clones. Hence, the total number of clone pairs within the benchmark is 58. Figure 4 shows an overview

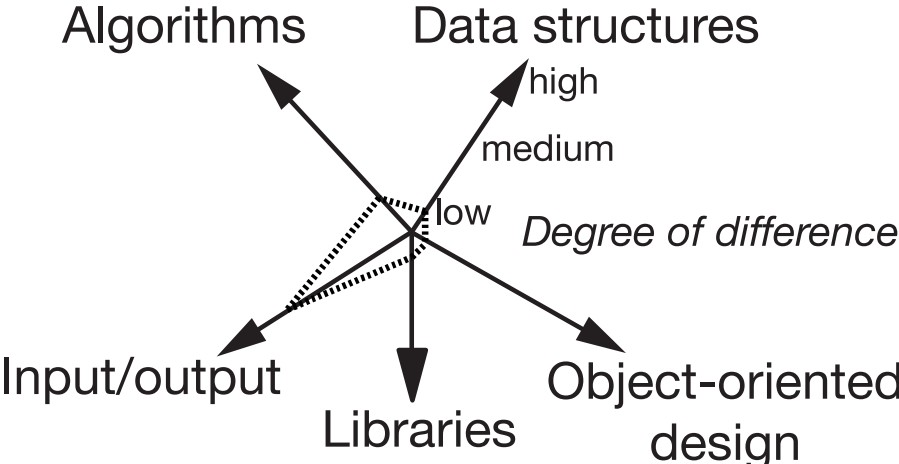

**Figure 2** Example category rating of a clone pair in the benchmark set.

```
public static void main(String[] args) {
  reader = new BufferedReader(
    new FileReader("A-large.in"));
  writer = new PrintWriter("a.out");
public static void main(String[] args) {
  File file = new File(System.in);
  try (Scanner scanner = new Scanner(
    new FileReader(file))) {
    File out = new File(System.out);
    try (PrintWriter writer = new ...
```

**Figure 3** Example of a high difference in the category Input/Output.

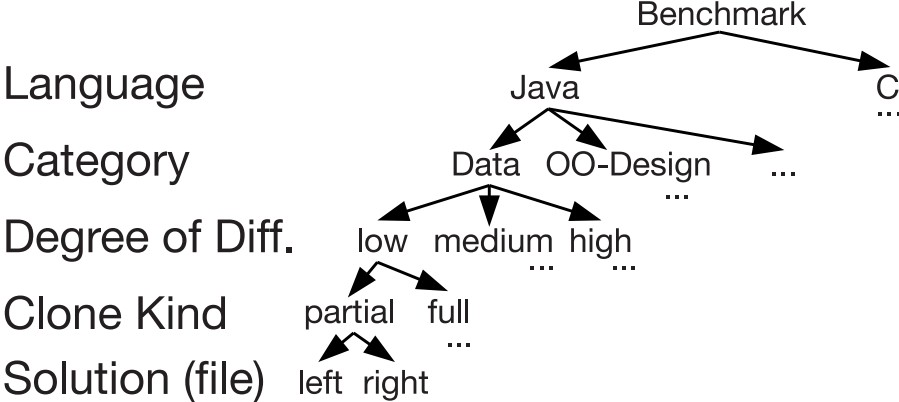

**Figure 4** Structure of the benchmark set (overview).

**Table 12** Kappa values for difference categories.

| Category | Kappa |
| --- | --- |
| Data structures | 0.41 |
| OO design | 0.35 |
| Algorithms | 0.47 |
| Libraries | 0.36 |
| Input/Output | 0.47 |

of the structure of the whole benchmark set. This structure enables developers of a clone detection tool to test their tool easily as well analyse the nature of the clones found and not found by a tool.

Our benchmark provides several advantages to the research community. First, *developers of a clone detection tool can easily test their tool with the source files as input.* They can see whether their tool detects the clones or they can analyse why it did not. Second, the clones in the benchmark are easily understandable examples for the categories we created. Third, *the clones in the benchmark were not built artificially; the solutions were implemented independently by at least two persons during the Code Jam contest.* Despite our modifications to avoid copyright problems, neither changing structure nor algorithm, the code clones are more realistic than fully artificial copies where one file is modified as part of a study.

# THREATS TO VALIDITY

We analyse the validity threats for this study following common guidelines for empirical studies (*Yin, 2003*; *Wohlin et al., 2012*).

## Conclusion validity

As most of our measurements and calculations were performed automatically, the threats to conclusion validity are low. For the corresponding hypothesis tests, we checked all necessary assumptions. Only the classification and rating of the degree of difference is done manually and, hence, could be unreliable. We worked in pairs to reduce this threat. Furthermore, one of the researchers performed an independent classification of a random sample of 10 clone pairs to compare the results. We calculated Cohen's kappa for the categories of differences between clone pairs as presented in Table 12.

We interpret the kappa results according to the classification by *Landis & Koch (1977)*. Hence, our results are a moderate agreement between the categories: data structures, algorithms and input/output. For the categories object-oriented design and libraries we have a fair agreement. We consider this to be reliable enough for our investigations.

## Internal validity

There is the threat that the implementation of our instrumentation tooling may contain faults and, therefore, compute incorrect results for the detected clones and recalls. We reduced this threat inherently by the manual inspections done to answer RQ 2 and independently to investigate the type-4 clones.

A further threat to internal validity is that we took our solution sets from Google Code Jam. We cannot be sure that all the published solutions of the Code Jam within a solution set are actually functionally similar. We rely on the fact that the organisers of the Code Jam must have checked the solutions to rank them. Furthermore, we assume to have noticed in the manual inspections if there were solutions in a solution set with highly differing functionality.

### Construct validity

To fully understand the effectiveness of a clone detection approach, we need to measure precision as well as recall. In our study, we could not measure precision directly because of the large sample size. We checked for false positives during the manual inspections and noted only few rather short clones. Our minimal clone length is below recommended thresholds. This is a conservative approach to the problem. By that, we will find more clones than in an industrial approach. We decided to use this threshold to be sure that we cover all the interesting clone pairs that would be lost due to variation in the precision of the tools.

There is a threat because we count each clone pair only once. In partial clones, one clone pair might contain a type-2 as well as a type-3 partial clone. In those cases, we decided that the lower type—the easier detection—should be recorded. Hence, the assignment to the types might be imprecise. We accept this threat as it has no major implication for the conclusions of the experiment.

### External validity

There is also a threat to external validity in our usage of solutions from Google Code Jam. The submitted programs might not represent industrial software very well. Participants had a time limit set for turning in their solutions. Furthermore, the programming problems contained mostly reading data, performing some calculations on it and writing data. This might impact the method structure within the solutions. This threat reduces the generalisability of our results. Yet, we expect that other, more complex software will introduce new kinds of difference categories (e.g., differences in GUI code) and only extend but not contradict our results.

For the study, we chose three well-known and stable clone detection tools. Two of them analyse Java and C programs detecting type-1 to type-3 clones. The third one detects type 4 clones and supports only programs written in C and only finds clones in complete functions. Overall, we are confident that these tools represent the available detection tools well.

## CONCLUSIONS AND FUTURE WORK

In this paper, we investigated the characteristics of clones not created by copy&paste. We base our study on *Juergens, Deissenboeck & Hummel (2010b)*, but this is the first study with programs implementing different specifications in diverse programming languages including CCCD as concolic clone detector for type-4 clones. We found that a full syntactic similarity was detected in less than 1% of clone pairs. Even partial syntactic similarity

was only visible in less than 12%. The concolic approach of CCCD can detect FSCs without syntactic similarity as type-4 clones. Yet, a full detection was only possible in 0.1% of clone pairs.

Our categorisation of the differences of clone pairs that were not syntactically similar showed that usually several characteristics make up these differences. On average, however, the differences were mostly small. Hence, we believe there is a huge opportunity to get a large improvement in the detection capabilities of type-4 detectors even with small improvements in tolerating additional differences. We provide a carefully selected benchmark with programs representing real FSCs. We hope it will help the research community to make these improvements.

### Relation to existing evidence

We can most directly relate our results to *Juergens, Deissenboeck & Hummel (2010b)*. We support their findings that using type-1–3 detectors, below 1% is fully and below 10% is partially detected. We can add that with the type-4 detection of CCCD, the partial clone recall can reach 16%. They introduce categories which were derived from other sources but not created them with a systematic qualitative analysis. Yet, there are similarities in the categories.

Their category *syntactic variation* covers "if different concrete syntax constructs are used to express equivalent abstract syntax." We categorised this as small algorithm difference. Their category *organisational variation* "occurs if the same algorithm is realized using different partitionings or hierarchies of statements or variables." We categorise these differences as a medium algorithm difference. Their category *delocalisation* "occurs since the order of statements that are independent of each other can vary arbitrarily between code fragments" is covered as difference in algorithm in our categorisation. Their category *generalisation* "comprises differences in the level of generalization" which we would cover under *object-oriented design*. They also introduce *unnecessary code* as category with the example of a debug statement. We did not come across such code in our sample but could see it as a potential addition.

Finally, they clump together *different data structure and algorithm* which we categorised into separate categories. We would categorise these variations as either data structure or algorithm differences with probably a high degree of difference. They found that 93% of their clone pairs had a variation in the category *different data structure or algorithm*. We cannot directly support this value but the tendency. We found that 91% of the inspected clone pairs had a difference at least in either *algorithm* or *data structure* and especially for algorithm the difference was on average large.

*Tiarks, Koschke & Falke (2011)* created a categorisation for differences in type-3 clones. Therefore, their focus was on classifying syntactic differences that probably hail from independent evolution of initially copied code. Yet, the larger the differences, the more their categories are similar to ours. For example, they abstract edit operations to *type substitution* or *different algorithms*. We believe, however, that our categorisation is more useful for FSCs and to improve clone detection tools along its lines.

### Impact

Independently developed FSCs have very little syntactic similarity. Therefore, type-1–3 clone detectors will not be able to find them. Newer approaches, such as CCCD, can find FSCs but their effectiveness still seems limited. Hence more research in approaches more independent of syntactic representations is necessary. We will need to find ways to transfer the positive results of *Jiang & Su (2009)* with the Linux kernel to other languages and environments while overcoming the challenges in such dynamic detections as discussed, for example, in *Deissenboeck et al. (2012)*. We hope our benchmark will contribute to this.

### Limitations

The major limitation of our study is that we did not use a wide variety of types of programs that exist in practice. The programs from Google Code Jam all solve structurally similar problems, for example, without any GUI code. We expect, however, that such further differences would rather decrease the syntactic similarity even more. The categories might have to be extended to cover these further differences. Nevertheless, the investigated programs were all developed by different programmers and are not artificial.

Furthermore, we had to concentrate on three clone detectors and two programming languages. Other tools and languages might change our results but we are confident that our selection is representative of a large class of detectors and programming languages.

### Future work

We plan to investigate the differences between the tools and the detected clone pairs of different types in more detail. In particular, we would like to work with researchers who have built type-4 detectors to test them against our clone database and to inspect the found and not found clones.

## ACKNOWLEDGEMENTS

The authors would like to thank Benjamin Hummel, Lingxiao Jiang and Daniel Krutz for their help in getting their tools to work and Kornelia Kuhle for feedback on the text.

### Funding

The authors received no funding for this work.

### Competing Interests

The authors declare there are no competing interests.

### Author Contributions

- Stefan Wagner conceived and designed the experiments, performed the experiments, analyzed the data, wrote the paper, prepared figures and/or tables, reviewed drafts of the paper.
- Asim Abdulkhaleq and Jasmin Ramadani performed the experiments, wrote the paper, prepared figures and/or tables, reviewed drafts of the paper.

- Ivan Bogicevic and Jan-Peter Ostberg performed the experiments, analyzed the data, contributed reagents/materials/analysis tools, wrote the paper, prepared figures and/or tables, performed the computation work, reviewed drafts of the paper.

### Data Availability

ZENODO: 10.5281/zenodo.12646.

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
