# Peer review of "How are functionally similar code clones syntactically different? An empirical study and a benchmark"

_PeerJ Computer Science, doi:10.7717/peerj-cs.49_

## Round 0.1 · original submission · Minor Revisions

This is an interesting paper about 'functionally similar clones': Pieces of code that exhibit (almost) the same behavior, but have been written by different developers, and hence may be structurally very different. The paper studies 2 * 1400 programs all submitted as solutions to the same Google Code Jam. It uses this large corpus to come up with:

1. The observation that functionally similar clones exhibit hardly any structural similarity (0% at full file level, < 16% at fragment level)
2. The observation that the only existing technique for detecting functionally similar clones is very limited (it found just one clone in 70 program pairs)
3. A categorization of typical syntactic differences between functionally similar clones (algorithm; data structure; OO; IO; libraries)
4. A smaller benchmark that can be used for further research into detection of functionally similar clones.

The underlying data set is available via GitHub, and published under DOI http://doi.org/10.5281/zenodo.12646.

Two reviewers are positive: They mainly have a number of clarification questions that should be easy to address.

The third reviewer refers to earlier feedback you received. My impression is that you have partially addressed this earlier feedback. Please double check the earlier feedback you received, and kindly indicate in your rebuttal how you addressed the main points.

Based on this, I share the reviewers' support for this paper. Future work on functional clones will either use the resulting benchmark, or build upon the findings in this paper.

I doubt whether the title does justice to the full paper. The question is not very clear (different from what?) and fails to clarify what the real contributions (such as the benchmark) of the paper are. Maybe you want to reconsider the title?

Details:
- Usually there should be no space before a "%".
- There were quite a overfull hboxes in the pdf. Please double check the formatting.
Intro:
- "hundreds of millions" is a *lot*. There are not that many systems that large, actually. Add a reference, or scale down maybe?
- I doubt Bellon et al were the first to introduce the type-1-4 terminology. Perhaps cite the original paper?

In short: If you handle the above minor points and clarify the paper to address the reviewers' concerns, this paper should be ready to go.

Reviewer 1 ·

Basic reporting

In the Intro. it is mentioned "traditional clone detection approaches and tools are hardly able to detect functionally equivalent clones...". but the reference does not cover the recent research in the past five years. Recently, there has been several work on Semantically similar clone detection by exploiting various representation of program such as AST, PDG, bytecode etc.

This sentence in Section 1.1 is not clear: "...make a functionally similar clone really diff erent from copy..". A FSC clone can be the result of copy&paste and then some immediate modification and customization. this kind of clones belong to which group?

Experimental design

The tools selected for the study are fine, but the justification in Section 3.4.1 is a little bit imprecise. For example the sentence that is supporting your choice is outdated (from 2010): "They have been described as most up-to-date implementations of token based and AST based clone detection algorithms". Or another case, "Commercial tools were not exact enough in what they detect."

Please report Number of methods in Table 3. Also, in Table 3, the size column is unnecessary.

Section 3.5.2, it is mentioned the parameters are set in a way to find more clones. However, min clone length == 6 seems too restrictive, specifically for small programs.

The most important aspect of Type-3 clone detection is the choice of similar threshold. Please, discuss them in Section 3.5.2.

Validity of the findings

"No Comments"

Additional comments

The paper reports an interesting analysis which looks into the source code of programming competitions. I liked the idea of analyzing such code repositories for clone detection research, and I believe this would be an interesting article for the community.

Reviewer 2 ·

Basic reporting

The paper is written clearly, very easy to read. It introduces an intriguing problem of detecting functionally similar code with sufficient background, and motives well why it is needed to study the characteristics of functionally similar code that cannot be detected as syntactically similar code. What is unclear though is the definition of "functionally similar code", in comparison with the definition of "functionally the same code"; probably better to clarify what is considered "similar", e.g., same input producing same output except for output formats, etc.

The contents are mostly self-contained except for more details about the tools and parameters used.

Although the paper says all the data are available in their github project, no RUL link is provided.

Experimental design

To study functionally similar code that cannot be detected as syntactically similar code, the paper chose to use code written in different programming languages from Google Code Jam Contest as the subjects for the study, which is a good choice. Code clone detection tools that can handle different programming languages are used so that the study can be carried out for different programming languages and different tools, which is a good design for achieving better generalizability.

However in the actual study, only two languages, Java and C, and 3 clone detection tools, ConQAT, Deckard, and CCCD, are considered; this limits the general applicability of the results, especially the hypotheses H1, H2, H4 which consider the differences between different languages and tools. May consider revising the hypotheses to be more specific with respect to the actual data used.

Validity of the findings

The paper aims to better understand the characteristics of functionally similar code that cannot be detected as syntactically similar code. With the limited dataset from Google Code Jam Contest, it categorizes the code differences between functional clone pairs into 5 categories, each of which is split into 3 levels of difference degrees. The results provide a new perspective on such differences, although the definition of "low", "medium", "high" of difference degrees are too vague and may not be easy to automate.

The benchmark containing categorized functional clones can be useful. However, the size of the benchmark seems to be too small in comparison with all the code from Google Code Jam Contest. Can the categorization of functional clones be automated, so as to make it more scalable to have larger benchmarks?

Additional comments

The paper addresses an important, but challenging problem. It provides a categorization of functionally similar code different from those categories in the literature, complementing other studies. Although the results showing the categories are interesting, it would be much better to have more insights and a more concrete future plan for designing a better functional clone detection technique. Just based on the empirical study results, it is still unclear to me how a better detection technique may be derived. So I think only the first part of the research objectives (as stated in Section 1.2: to better understand the differences...to support future research on their detection) is achieved.

Minor issues about writing:
- Page 10, "similar CVS Files" -> CSV
- Page 14, "The other categories of the pairs are very low, ideally zero." I don't understand the rationale of the sentence.
- Page 14, "A final set of clone pairs that are not detected as full clones..." Does this mean the benchmark include functional clones that can be detected partial syntactic clones? Why don't exclude partial clones too to make the benchmark "stronger"?
- Page 22, among the 3 advantages mentioned, only the second one may be considered as a contribution of this paper; may make this point clear when comparing to other studies in the literature.

Reviewer 3 ·

Basic reporting

See general comments.

Experimental design

See general comments.

Validity of the findings

See general comments.

Additional comments

The presented submission is almost identical to a previous submission to a conference for which I have been a reviewer. The submission was rejected at the conference and the authors received detailed and valuable feedback. However, the current submission has not considered the feedback and the only changes are the addition of a few tables and four paragraphs of text.

I appreciate the effort that the authors have invested in the paper and believe that in the end, the work can lead to publishable material after the authors have considered the comments they have received for their earlier version.

As the paper has not been significantly changed compared to the conference version, I refer the authors to the comments they have received for that version.

---

## Round 0.2 · accepted · Accept

You carefully addressed all reviewer comments in a detailed response letter. This has resulted in a number of small yet valuable improvements to the paper. I consider the resulting paper as a strong contribution to the field of clone detection, and I am pleased to recommend it for acceptance.